# Bounds on the Lifetime Expectations of Series Systems with IFR Component Lifetimes

**DOI:** 10.3390/e23040385

**Published:** 2021-03-24

**Authors:** Tomasz Rychlik, Magdalena Szymkowiak

**Affiliations:** 1Institute of Mathematics, Polish Academy of Sciences, Śniadeckich 8, 00656 Warsaw, Poland; 2Institute of Automation and Robotics, Poznan University of Technology, Plac Marii Skłodowskiej-Curie 5, 60965 Poznan, Poland; magdalena.szymkowiak@put.poznan.pl

**Keywords:** series system, increasing failure rate, increasing density, convex transform order, generalized cumulative residual entropy, expectation, upper bound

## Abstract

We consider series systems built of components which have independent identically distributed (iid) lifetimes with an increasing failure rate (IFR). We determine sharp upper bounds for the expectations of the system lifetimes expressed in terms of the mean, and various scale units based on absolute central moments of component lifetimes. We further establish analogous bounds under a more stringent assumption that the component lifetimes have an increasing density (ID) function. We also indicate the relationship between the IFR property of the components and the generalized cumulative residual entropy of the series system lifetime.

## 1. Introduction

The series system composed o *n* items has the structure function $\varphi :{\left\{0,1\right\}}^{n}\mapsto \left\{0,1\right\}$ defined as $\varphi \left(x_{1},\ldots ,x_{n}\right)=\min\left\{x_{1},\ldots ,x_{n}\right\}$. If $X_{1},\ldots ,X_{n}$ denote the random lifetimes of the components, then the system lifetime is $X_{1:n}=\min\left\{X_{1},\ldots ,X_{n}\right\}$, where $X_{1:n},\ldots ,X_{n:n}$ denote the order statistics of $X_{1},\ldots ,X_{n}$. The series systems are the most important objects of the reliability theory. They are composed of the minimal number of components necessary for system operating. The failure of any component results immediately in the failure of the whole system. Series structures occur often as submodules of complex systems and networks. Applications of the series systems go far beyond the reliability of technical devices. They are useful, e.g., in economy (see, Ge et al. [1]), chemistry (see, Showalter and Epstein [2]) and industry (see, Xie and Gong [3]). Many survival analyses were based on reliability models of competing risks such that the first failure from a list of risks (e.g., cancer, heart disease, accident, etc.) causes the system failure (see, Kvam and Singh [4]). Furthermore, series systems are utilized by genetic algorithms and evolutionary theory (see, e.g., Jäntschi and Bolboacă [5]). In the paper, we consider the series systems whose components have independent and identically distributed (iid, for short) lifetimes with a common increasing failure rate (IFR, for short).

An absolutely continuous nonnegative life distribution function *F* with the density function *f* is called the increasing failure rate distribution function if the respective failure rate function $\lambda \left(x\right)=\frac{f(x)}{1-F(x)}$, x>0, is nondecreasing. The name is inherited by random variables which have IFR distribution functions. The increasing failure rate is the most classic notion used for describing a natural increasing fatigue tendency of appliances and their components during operation. The wearing tendency of the components (and the system, in consequence) is the effect of increasing fatigue of working components in the operating time, and increased burden caused by possible failures of other components. Another useful definition of the IFR property is based on the convex transform relation of the distribution with the standard exponential distribution Z(x)=1−exp(−x),x>0. The order was introduced by van Zwet [6]. We say that *F* is the IFR distribution if it precedes the exponential distribution in the convex transform order, i.e., the composition $F^{-1}\circ Z\left(x\right)$ is concave on $R_{+}$. This definition of IFR distributions generalizes the the traditional one in two directions. Firstly, by the location-scale invariance of the convex transform order, the support of *F* does not need to be restricted to the positive half-axis. Secondly, the convex transform order definition admits distribution functions which have atoms at the right-ends of their supports. The IFR properties of the series systems, and their dual properties of decreasing failure rate (DFR, for short) were studied by many researchers, e.g., by Barlow and Proschan [7], Lai and Xie [8], and Hazra et al. [9].

A more stringent concept of aging is expressed by the increasing density property (we use further the acronym ID for brevity). It is obvious that if the density function f(x) is nondecreasing on the support of *F*, then so is the product $\lambda \left(x\right)=f\left(x\right)\frac{1}{1-F(x)}$. The ID property may be also defined with use of the convex transform relation. A distribution function *F* has the ID property if it precedes the standard uniform distribution function U(x)=x, 0<x<1, in the convex transform order, which means that $F^{-1}\circ U\left(x\right)=F^{-1}\left(x\right)$ is concave on (0,1). This defines the family of distributions with nondecreasing density functions on some interval, and possible atom at its right end. This definition is also more restrictive than the IFR property because *Z* precedes *U* in the convex transform order, and the order is transitive.

Recently, various versions of entropy were applied for measuring uncertainty of random life data, and order statistics and reliability system lifetimes in particular. Recall that for lifetime random variable *X* supported on [0,∞) the classic entropy of Shannon [10] is defined as
$$H\left(X\right)=-\int_{0}^{\infty }f\left(x\right)\ln f\left(x\right)dx.$$

Rao et al. [11] defined and studied the cumulative residual entropy
$$E\left(X\right)=-\int_{0}^{\infty }\left(1-F(x)\right)\ln\left(1-F(x)\right)dx,$$
and the generalized cumulative residual entropy
$$E_{n}\left(X\right)=\int_{0}^{\infty }\left(1-F(x)\right)\frac{{\left[-\ln\left(1-F(x)\right)\right]}^{n}}{n!}dx$$
was proposed in Psarrakos and Navarro ([12]. For instance, Baratpour [13] provided conditions for characterization of the component lifetime distribution by the cumulative residual entropy of the series system lifetime. Sunoj et al. [14] analyzed the quantile-based entropy of order statistics. Toomaj and Di Crescenzo [15] presented the relationships between the generalized cumulative residual entropy and monotone failure rate properties. They proved, e.g., that if components of the series system are iid and belong to the IFR family, then the generalized cumulative residual entropy of the conditional residual system lifetime $E_{n}\left(X_{1:n}-t|X_{1:n}>t\right)$, t>0, is decreasing. Other connections between Shannon entropy (see Wang et al. [16], Jäntschi and Bolboacă [17], Tănăsescu and Popescu [18]) and order statistics were presented, e.g., by Koné [19], Jäntschi [20], Courtier et al. [21] and Cofré et al. [22].

We assume here that the common marginal component lifetime distribution function *F* belongs either to the IFR or ID family and determine sharp upper bounds on the standardized expectations $\frac{EX_{1:n}-\mu }{\sigma_{p}}$ of series system lifetime, where $\mu =EX_{1}$ is the expectation of the parent distribution function *F*, and $\sigma_{p}={\left({E|X-\mu |}^{p}\right)}^{\frac{1}{p}}$, p≥1, is the *p*th root of the respective *p*th absolute central moment. Writing $\sigma_{p}$, we tacitly assume that this is positive and finite, i.e., *F* is nondegenerate and it has a finite *p*th moment. Observe that our bounds are necessarily nonpositive, because $EX_{1:n}<\mu$ for every iid sample with a nondegenerate baseline distribution function *F*. In this paper, we show that for every parent distribution with increasing density and failure rate functions, and every p>1 the bounds for standardized expectations of series system lifetimes trivially amount to 0. Farther, we determine strictly negative bounds expressed the scale units $\sigma_{1}$, being the absolute mean deviation from the population mean. Obviously, the bounds for the IFR family occur greater than those in the ID case, because the IFR family is substantially greater than the class of increasing density distributions. We also describe the parent distributions which attain the bounds. They are elements of the families preceding the exponential and uniform distributions in the convex transform order that, in some cases, possibly have atoms at the right end of their supports. However, these extreme distributions can be approximated by sequences of absolutely continuous members of the IFR and ID classes, having even strictly increasing failure rate and density functions, respectively. Hence the bounds are attained in the limit by these sequences.

Obviously, the series systems are the particular cases (with k=n) of the *k*-out-of-*n* systems, k=1,…,n, that work as long as do so at least *k* of its *n* components. The mean-standard deviation bounds on the expectations of *k*-out-of *n* systems with k=1,n−1,n (called the parallel, fail-safe, and series systems, respectively) when the component lifetimes are iid and belong to the ID and IFR families, were determined by Rychlik [23]. However, in the case of series systems only the trivial zero bounds were established there. Analogous results for the other *k*-out-of-*n* systems were presented by Goroncy and Rychlik in [24] for the ID case, and in [25] for the IFR family. The evaluations for the dual DD and DFR families were determined several years earlier. In particular, Danielak [26] presented positive mean-standard deviations bounds on the expectations of order statistics $X_{k:n}$ with ranks *k* relatively large with respect sample sizes *n* coming from populations with decreasing density and failure rate functions. These correspond to the *k*-out-of-*n* systems with *k* relatively far from *n*. Rychlik [27,28] established negative sharp bounds on the differences $E(X_{k:n}-X_{1})$ for small rank order statistics from the DD and DFR families, respectively, expressed in various scale units. As we can conclude from the above short review, there are known the sharp bounds on the expected lifetimes of all *k*-our-of-*n* systems under various assumptions on monotonicity of the failure rate and density function of lifetimes of their identical components except for the series systems under the IFR and ID assumptions. In the paper, we fill this important gap.

The rest of our paper is organized as follows. In Section 2 we show that the obvious zero upper bounds for the differences between the expectations of series system and its single component lifetimes cannot be improved in the ID and IFR cases if the differences are gauged in the scale units based on central absolute moments of orders greater than 1. In Section 3 we obtain refined nontrivial strictly negative evaluations of the differences expressed in the mean absolute deviation units. It occurs that the bounds under the more restrictive assumption that the distributions of component lifetimes have a common increasing density function are substantially tighter than in the case that we only assume that the distributions have merely an increasing failure rate. Section 4 contains a brief summary of the results established in the paper.

## 2. Zero Bounds

In contrast with the DD and DFR populations, the nonpositive upper bounds on the differences between the expectations of order statistics and population means in the ID and IFR cases are possible only for the sample minima which in reliability problems represent the lifetimes of series systems. Therefore it is of a vital interest to evaluate the expectations of the system lifetimes, especially under the practically important assumption that the component lifetime distributions have the IFR property.

**Theorem** **1** (ID and IFR distributions)**.***Let*$X_{1},\ldots ,X_{n}$*be iid random variables with a convex baseline distribution function and finite pth absolute central moment*$\sigma_{p}^{p}$*for some*p>1. *Then the following bound*(1)$$\frac{EX_{1:n}-\mu }{\sigma_{p}}\leq 0,$$*is sharp, and the equality is attained there in the limit by the sequences of mixtures of uniform distributions with atoms at the right ends of their support, when the contributions of the absolute continuous parts vanish.*

Obviously, the mixtures described in the theorem have convex distribution functions. Since each convex distribution function has a convex cumulative failure rate, and $X_{1:n}\leq X_{1}$ for all parent distributions, then the bound (Equation 1) is valid and sharp for the family of IFR distributions as well. The statement of Theorem 1 was determined in Rychlik [23] in the special case p=2.

**Proof.** Consider first the family of mixtures of uniform distributions on [0,1], and atoms at 1 with probabilities 0<α<1 and 1−α, respectively. They have distribution functions
$$F_{\alpha }\left(x\right)=\left\{\begin{array}{cc} 0, & x\leq 0,\\ \alpha x, & 0\leq x<1,\\ 1, & x\geq 1, \end{array}\right.$$
which are convex on their common support [0,1], and concave quantile functions
(2)$$F_{\alpha }^{-1}\left(x\right)=\left\{\begin{array}{cc} \frac{x}{\alpha }, & 0<x\leq \alpha ,\\ 1, & \alpha \leq x<1. \end{array}\right.$$
We easily check that for iid $Y_{1},\ldots ,Y_{n}$ with the above parent distributions we have
(3)$$\begin{array}{ccc} E_{\alpha }Y_{1} & = & \int_{0}^{\infty }\left[1-F_{\alpha }\left(x\right)\right]dx=\int_{0}^{1}\left(1-\alpha x\right)dx=1-\frac{\alpha }{2}, \end{array}$$
(4)$$\begin{array}{ccc} E_{\alpha }Y_{1:n} & = & \int_{0}^{\infty }{\left[1-F_{\alpha }\left(x\right)\right]}^{n}dx=\int_{0}^{1}{\left(1-\alpha x\right)}^{n}dx=\frac{1-{\left(1-\alpha \right)}^{n+1}}{(n+1)\alpha } \end{array}$$
so that
(5)EαY1:n−EαY1=1−(n+1)α+n+12α2−(1−α)n+1(n+1)α=−n−12α+∑j=2nnn−1j−1(−1)jαj.
The respective *p*th absolute central moments are
(6)$$\begin{array}{ccc} N_{p}^{p}\left(\alpha \right) & = & \alpha \int_{0}^{1}{\left|x-1+\frac{\alpha }{2}\right|}^{p}dx+\left(1-\alpha \right)\frac{\alpha^{p}}{2^{p}}\\ & = & \alpha \;\frac{{\left(2-\alpha \right)}^{p+1}+\alpha^{p+1}+2\left(p+1\right)\left(1-\alpha \right)\alpha^{p-1}}{2^{p+1}\left(p+1\right)}. \end{array}$$
For p>1 we have
(7)$$\lim_{\alpha \to 0}{\left(\frac{N_{p}^{p}\left(\alpha \right)}{\alpha }\right)}^{\frac{1}{p}}=\frac{1}{2^{p+1}\left(p+1\right)}.$$
Combining (Equation 5) with (Equation 7) we obtain
$$\lim_{\alpha \to 0}\frac{E_{\alpha }Y_{1:n}-E_{\alpha }Y_{1}}{N_{p}\left(\alpha \right)}=\lim_{\alpha \to 0}-\left(n-1\right)2^{p}\left(p+1\right)\alpha^{1-1/p}=0.$$
Obviously, replacing our $Y_{i}$ by $X_{i}=\mu +\sigma_{p}\;\frac{Y_{1}-1+\frac{\alpha }{2}}{N_{p}\left(\alpha \right)}$, i=1,…,n, which have arbitrarily chosen means μ∈R, *p*th absolute central moments $\sigma_{p}^{p}>0$, and convex distribution functions, we attain the zero bound on $\frac{EX_{1:n}-\mu }{\sigma_{p}}$ when α→0 as well. □

## 3. Negative Bounds

The results of the previous section show that we get noninformative zero bounds on the differences between the expectations of the system and component lifetimes if we measure the differences in non-sufficiently subtle scale units, including the standard deviation of the component lifetime in particular. The conclusions essentially change if we replace the scale units $\sigma_{p}$, p>1, by the mean absolute deviation from the mean $\sigma_{1}=E\left|X_{1}-\mu \right|$. We first consider the series systems whose components have IFR lifetimes.

**Theorem** **2** (IFR distributions)**.***For the minimum of n iid random variables with a common distribution that has the convex cumulative failure rate and finite mean μ and first absolute central moment*$\sigma_{1}$, *the following inequality*(8)$$\frac{EX_{1:n}-\mu }{\sigma_{1}}\leq -\frac{n-1}{n}\;\frac{e}{2}\;,$$*holds. It becomes equality in the limit for the mixtures of exponential distributions with location parameter (left-end support point)*$\mu -\sigma_{1}\frac{1-e^{-\alpha }}{N(\alpha )}$*and intensity α truncated right at*$\mu +\sigma_{1}\frac{e^{-\alpha }}{N(\alpha )}$, *and atom at*$\mu +\sigma_{1}\frac{e^{-\alpha }}{N(\alpha )}$*with probabilities*$1-e^{-\alpha }$ and $e^{-\alpha }$
*, respectively, when*
α→∞
*, where*
(9)$$N\left(\alpha \right)=\frac{1}{\alpha }\left[2\exp\left(-e^{-\alpha }-1\right)-e^{-\alpha }-e^{-2\alpha }-\alpha e^{-\alpha }\right].$$


**Proof.** The idea of proof is based on the norm maximization method proposed by Goroncy and Rychlik [29] (see also Rychlik [27,28], and Goroncy and Rychlik [25]).Let $f_{1:n}\left(x\right)=n{\left(1-x\right)}^{n-1}$, 0<x<1, denote the density function of the first order statistic of the standard uniform sample of size *n*. Let $Z\left(x\right)=1-e^{-x}$, x>0, be the standard exponential distribution function. We first represent
(10)$$\begin{array}{ccc} E\frac{X_{1:n}-\mu }{\sigma_{1}} & = & \int_{0}^{1}\frac{F^{-1}\left(x\right)-\mu }{\sigma_{1}}\;f_{1:n}\left(x\right)dx=\int_{0}^{1}\frac{F^{-1}\left(x\right)-\mu }{\sigma_{1}}\left[f_{1:n}\left(x\right)-1\right]dx\\ & = & \int_{0}^{\infty }\frac{F^{-1}\circ Z\left(x\right)-\mu }{\sigma_{1}}\left[f_{1:n}\circ Z\left(x\right)-1\right]e^{-x}\;dx \end{array}$$
as a linear functional
$$T_{h}\left(g\right)=\int_{0}^{\infty }g\left(x\right)h\left(x\right)e^{-x}\;dx$$
on the elements of the Banach space $L^{1}\left(R_{+},e^{-x}\;dx\right)$. Here
$$h\left(x\right)=f_{1:n}\circ Z\left(x\right)-1=ne^{-(n-1)x}-1,$$
and
(11)$$g\left(x\right)=\frac{F^{-1}\circ Z\left(x\right)-\mu }{\sigma_{1}}$$
is an arbitrary element of the family
(12)$$\begin{array}{ccc} S_{1} & = & \{g\in L^{1}\left(R_{+},e^{-x}\;dx\right):g-\mathrm{nondecreasing}\;\mathrm{concave},\\ & & \;\;\;\;\;T_{1}\left(g\right)={0,\;||g||}_{1}=1\}, \end{array}$$
where
(13)$$T_{1}\left(g\right)=\int_{0}^{\infty }g\left(x\right)e^{-x}\;dx.$$
Indeed, each $g\in S_{1}$ is nondecreasing, because this is an increasing linear transformation of a quantile function composed with the exponential distribution function, and concave, by the IFR property of *F*. Vanishing of (Equation 13) for *g* from (Equation 12) follows the representation (Equation 11) and the fact that
$$\mu =\int_{0}^{1}F^{-1}\left(x\right)\;dx=\int_{0}^{\infty }F^{-1}\circ Z\left(x\right)e^{-x}\;dx.$$
Similarly, (Equation 11) and
$$\sigma_{1}=\int_{0}^{1}|F^{-1}\left(x\right)-\mu |\;dx=\int_{0}^{\infty }|F^{-1}\circ Z\left(x\right)-\mu |e^{-x}\;dx=\left|\right|F^{-1}\circ Z-\mu {\left|\right|}_{1}$$
imply ${\left||g|\right|}_{1}=1$.A crucial observation now is that
$$T_{h}\left(g\right)<0,\;g\in S_{1},$$
because $EX_{1:n}<EX_{1}$ for all nondegenerate iid random variables with a finite first moment (equivalently, with positive finite $\sigma_{1}$). This allows us to use the norm maximization method. Noting that the problem of maximizing (Equation 10) is location and scale invariant, we can replace the original problem by the dual one of maximizing ${\left||g|\right|}_{1}$ over the set
(14)$$\begin{array}{ccc} S_{2} & = & \{g\in L^{1}\left(R_{+},e^{-x}\;dx\right):g-\mathrm{nondecreasing}\;\mathrm{concave},\\ & & \;\;\;\;\;T_{1}\left(g\right)=0,\;T_{h}\left(g\right)=-1\}. \end{array}$$
Indeed, replacing $g\in S_{1}$ by $\tilde{g}=\frac{g}{-T_{h}\left(g\right)}$, we get $T_{h}\left(\tilde{g}\right)=-1$ and $\left|\right|\tilde{g}{\left|\right|}_{1}=\frac{1}{|T_{h}\left(g\right)|}=-\frac{1}{T_{h}\left(g\right)}$. It means that maximizing the norm over (Equation 14) we also maximize $T_{h}\left(g\right)$ over the original set (Equation 12).The next step of our reasoning is based on the observations that (Equation 14) is a convex set, and the norm functional is convex. It follows that for every convex combination of elements of (Equation 14) we have
$${∥\sum_{i=1}^{k}\gamma_{i}g_{i}∥}_{1}\leq \sum_{i=1}^{k}\gamma_{i}{∥g_{i}∥}_{1}\leq \max_{1\leq i\leq k}{∥g_{i}∥}_{1},$$
and in consequence, as k→∞,
$${∥\sum_{i=1}^{\infty }\gamma_{i}g_{i}∥}_{1}\leq \underset{i\in N}{\sup}{∥g_{i}∥}_{1}.$$Now we notice that every $g\in S_{2}$ can be approximated by convex combinations of the elements of a particular subset of $S_{2}$ consisting of simple broken lines
(15)$$g_{\alpha }\left(x\right)=b\left(\alpha \right)\left(x-\alpha \right)1_{\left[0,\alpha \right]}\left(x\right)+a\left(\alpha \right),\;0\leq \alpha <\infty .$$
The slopes b(α)>0 and intercepts a(α) are uniquely determined by the equations $T_{1}\left(g_{\alpha }\right)=0$ and $T_{h}\left(g_{\alpha }\right)=-1$, but we do not need here their precise forms. The approximation procedure is standard, we restrict ourselves to outlining the main steps, and do not perform all the rigorous calculations. We first observe that each piecewise linear element of (Equation 14) is a convex combinations of some broken lines (Equation 15). The breaking points uniquely determine the elements of the combination, and the values of consecutive slopes allow us to calculate the respective mixture weights. Then we observe that each $g\in S_{2}$ can be linearly interpolated by sequences of piecewise linear, continuous, nondecreasing, and concave functions $g_{k}$, k→∞, with the increasing sets of knots which in the limit become dense in $R_{+}$. These approximations $g_{k}$ do not necessarily belong to $S_{2}$, but they tend monotonously to *g* on the whole positive half-axis, which implies that
$$\begin{array}{ccc} T_{1}\left(g_{k}\right) & \to & T_{1}\left(g\right)=0,\\ T_{h}\left(g_{k}\right) & \to & T_{h}\left(g\right)=-1, \end{array}$$
as k→∞. However, the following modifications
$${\tilde{g}}_{k}=\frac{g_{k}-T_{1}\left(g_{k}\right)}{|T_{h}\left(g_{k}\right)|},\;k=1,2,\ldots$$
do belong to (Equation 14), and they satisfy
$$\begin{array}{ccc} & & \left|\right|{\tilde{g}}_{k}-g{\left|\right|}_{1}={\left|\left|\frac{g_{k}}{|T_{h}\left(g_{k}\right)|}-g_{k}+g_{k}-g-\frac{T_{1}\left(g_{k}\right)}{|T_{h}\left(g_{k}\right)|}1_{R_{+}}\right|\right|}_{1}\\ & \geq & \left|\frac{1}{|T_{h}\left(g_{k}\right)|}-1\right|\left|\right|g_{k}{\left|\right|}_{1}+\left|\right|g_{k}-g{\left|\right|}_{1}+\left|\frac{T_{1}\left(g_{k}\right)}{T_{h}\left(g_{k}\right)}\right|{\left|\left|1_{R_{+}}\right|\right|}_{1}\to 0, \end{array}$$
because $\left|\right|g_{k}{\left|\right|}_{1}\leq {\left||g|\right|}_{1}<\infty$, the last norm amounts to 1, and the remaining three terms in the last line vanish as k→∞. It follows that $\left|\right|{\tilde{g}}_{k}{\left|\right|}_{1}\to {\left||g|\right|}_{1}$ by continuity of the norm functional. Therefore
$${\left||g|\right|}_{1}\leftarrow \left|\right|{\tilde{g}}_{k}{\left|\right|}_{1}={\left|\left|\sum_{i=1}^{k}\gamma_{i}g_{\alpha_{i}}\right|\right|}_{1}\max_{1\leq i\leq k}\left|\right|g_{\alpha_{i}}{\left|\right|}_{1},$$
and finally
$${\left||g|\right|}_{1}\leq \underset{0<\alpha <\infty }{\sup}\left|\right|g_{\alpha }{\left|\right|}_{1},\;g\in S_{2},$$
which implies that in our dual maximization problem (and the original one as well) we can restrict ourselves to considering first increasing and then constant broken lines (Equation 15) with varying breaking points.Referring again to location-scale invariance of the problem, we consider the simplest representatives of the location-scale families. Precisely, we confine our attention to the parent distribution functions $F_{\alpha }$, α>0, satisfying
$$F_{\alpha }^{-1}\left(1-e^{-x}\right)=\left\{\begin{array}{cc} \frac{x}{\alpha }, & 0<x\leq \alpha ,\\ 1, & x\geq \alpha , \end{array}\right.$$
(cf. (Equation 2)). The explicit forms of the distribution functions are
(16)$$F_{\alpha }\left(x\right)=\left\{\begin{array}{cc} 0, & x\leq 0,\\ 1-e^{-\alpha x}, & 0\leq x<1,\\ 1, & x\geq 1, \end{array}\right.$$
which means that they have absolutely continuous parts with the density functions $f_{\alpha }\left(x\right)=\alpha \exp\left(-\alpha x\right)$ on [0,1], and jumps of heights $e^{-\alpha }$ at 1.If $Y_{1},\ldots ,Y_{n}$ are iid with the distribution function $F_{\alpha }$, then
$$\begin{array}{ccc} E_{\alpha }Y_{1} & = & \int_{0}^{1}e^{-\alpha x}\;dx=\frac{1-e^{-\alpha }}{\alpha },\\ E_{\alpha }Y_{1:n} & = & \int_{0}^{1}e^{-n\alpha x}\;dx=\frac{1-e^{-n\alpha }}{n\alpha }, \end{array}$$
and
$$E_{\alpha }\left(Y_{1:n}-Y_{1}\right)=\frac{1}{n\alpha }\left(1-n+ne^{-\alpha }-e^{-n\alpha }\right).$$
We also determine
$$E_{\alpha }\left|Y_{1}-E_{\alpha }Y_{1}\right|=\alpha \int_{0}^{1}\left|x-\frac{1-e^{-\alpha }}{\alpha }\right|e^{-x}dx+\left(1-\frac{1-e^{-\alpha }}{\alpha }\right)e^{-\alpha }=N\left(\alpha \right).$$(see (Equation 9)). Accordingly,
(17)$$\frac{E_{\alpha }\left(Y_{1:n}-Y_{1}\right)}{E_{\alpha }\left|Y_{1}-E_{\alpha }Y_{1}\right|}=\frac{1}{n}\;\frac{1-n+ne^{-\alpha }-e^{-n\alpha }}{2\exp\left(-e^{-\alpha }-1\right)-e^{-\alpha }-e^{-2\alpha }-\alpha e^{-\alpha }}=B_{n}\left(\alpha \right),$$
say. Our goal is to maximize each $B_{n}\left(\alpha \right)$, n≥2, with respect to 0<α<∞.Fix first n=2 and change the variable for $x=1-e^{-\alpha }\in \left(0,1\right)$ in (Equation 17). Then we obtain
$$2B_{2}\left(-\ln\left(1-x\right)\right)=\frac{-x^{2}}{2e^{-x}-2+3x-x^{2}+\left(1-x\right)\ln\left(1-x\right)}=\frac{-x^{2}}{D(x)}.$$
We develop the denominator into the power series
$$\begin{array}{ccc} D(x) & = & 2\sum_{i=0}^{\infty }\frac{{\left(-1\right)}^{i}}{i!}x^{i}-2+3x-x^{2}-\left(1-x\right)\sum_{i=1}^{\infty }\frac{1}{i}x^{i}\\ & = & \frac{x^{2}}{2}+\sum_{i=3}^{\infty }\left[\frac{2{\left(-1\right)}^{i}}{i!}-\frac{1}{i}+\frac{1}{i-1}\right]x^{i}. \end{array}$$
Therefore
(18)$$g_{2}\left(x\right)=\frac{-1}{2B_{2}\left(-\ln\left(1-x\right)\right)}=\frac{D(x)}{x^{2}}=\frac{1}{2}+\sum_{i=1}^{\infty }\frac{1}{\left(i+1)(i+2\right)}\left[1+\frac{2{\left(-1\right)}^{i}}{i!}\right]x^{i}.$$
The derivative has the following expansion
(19)$$g_{2}^{\prime }\left(x\right)=\sum_{i=0}^{\infty }\frac{i+1}{\left(i+2)(i+3\right)}\left[1-\frac{2{\left(-1\right)}^{i}}{(i+1)!}\right]x^{i}=\sum_{i=0}^{\infty }a_{i}x^{i},$$
where $a_{0}=-\frac{1}{6}<0$, $a_{1}=\frac{1}{3}>0$, and $a_{i}>0$, i=2,3,…, as well, because $\left|\frac{2{\left(-1\right)}^{i}}{(i+1)!}\right|<1$ for all i≥2. This means that $g_{2}^{\prime }\left(0\right)<0$ and $g_{2}^{\prime }$ increases on (0,1), where it is well defined. It follows that $g_{2}$ either always decreases on (0,1) or it first decreases and then increases there. Anyway, the global maximum is attained at either of the end-points. Consequently,
$$B_{2}\left(\alpha \right)=\frac{-1}{2g_{2}(1-e^{-\alpha )}}$$
attains its global supremum either at 0 or at +∞, because functions $\left(0,\infty \right)∋\alpha \mapsto x=1-e^{-\alpha }\in \left(0,1\right)$ and $\left(0,1\right)∋x\mapsto \frac{-1}{2x}\in \left(-\infty ,-\frac{1}{2}\right)$ are increasing. We easily check that
$$\lim_{\alpha \to \infty }B_{2}\left(\alpha \right)=-\frac{e}{4}>\lim_{\alpha \to 0}B_{2}\left(\alpha \right)=-1.$$Using the transformation $x=1-e^{-\alpha }$ for n=3 we get
$$B_{3}\left(-\ln\left(1-x\right)\right)=\frac{2}{3}\;\left(3-x\right)B_{2}\left(x\right)=-\frac{1}{3}\;\frac{3-x}{g_{2}\left(x\right)}.$$
We verify variability of $g_{3}\left(x\right)=\frac{g_{2}\left(x\right)}{3-x}$ analyzing the sign of
$$g_{3}^{\prime }\left(x\right){\left(3-x\right)}^{2}=g_{2}^{\prime }\left(x\right)\left(3-x\right)+g_{2}\left(x\right).$$
Applying (Equation 18) and (Equation 19), by elementary calculations we obtain the Taylor expansion of
$$g_{3}^{\prime }\left(x\right){\left(3-x\right)}^{2}=\sum_{i=1}^{\infty }\left(b_{i}+c_{i}\right)x^{i},$$
where
$$b_{i}=\frac{3(i+1)}{\left(i+2)(i+3\right)}-\frac{i+2}{\left(i+3)(i+4\right)}+\frac{1}{\left(i+1)(i+2\right)}=\frac{2i^{3}+14i^{2}+26i+20}{\left(i\;+\;1)(i\;+\;2)(i\;+\;3)(i\;+\;4\right)}$$
are positive, and
$$c_{i}=\frac{2{\left(-1\right)}^{i}}{(i+4)!}\left[\left(i+3\right)\left(i+4\right)-3\left(i+1\right)\left(i+4\right)-i-2\right]=\frac{2{\left(-1\right)}^{i+1}}{(i+4)!}\left(2i^{2}+9i+2\right)$$
change the sign. We easily check that
$$\frac{|c_{i}|}{b_{i}}=\frac{2i^{2}+9i+2}{i^{3}+7i^{3}+13i+10}\;\frac{1}{i!}<\frac{7i^{3}+13i+10}{i^{3}+7i^{3}+13i+10}\;\frac{1}{i!}\leq 1,$$
which implies that $b_{i}+c_{i}>0$, i=1,2,…, and so the derivative of $g_{3}$ is positive. Accordingly,
$$B_{3}\left(\alpha \right)=-\frac{1}{3}\;\frac{1}{g_{3}\left(1-e^{-\alpha }\right)}$$
is increasing.We apply the claim for establishing the same conclusion for n≥4. Consider the functions
$$M_{n}\left(x\right)=x^{n}-nx+n-1,\;0<x<1,\;n\geq 3,$$
with the derivatives
$$M_{n}^{\prime }\left(x\right)=n\left(x^{n-1}-1\right),$$
and analyze variability of $\frac{M_{n+1}\left(x\right)}{M_{n}\left(x\right)}$, n≥3. We have
(20)$$\begin{array}{ccc} M_{n}^{2}\left(x\right)\;\frac{d}{dx}\;\frac{M_{n+1}\left(x\right)}{M_{n}\left(x\right)} & = & M_{n+1}^{\prime }\left(x\right)M_{n}\left(x\right)-M_{n+1}\left(x\right)M_{n}^{\prime }\left(x\right)\\ & = & \left(n+1\right)\left(x^{n}-1\right)\left[x^{n}-nx+n-1\right]\\ & - & n\left(x^{n-1}-1\right)\left[x^{n+1}-\left(n+1\right)x+n\right]\\ & = & x^{2n}-n^{2}x^{n+1}+2\left(n^{2}-1\right)x^{n}-n^{2}x^{n-1}+1\\ & = & x^{2n}-2x^{n}+1-n^{2}x^{n-1}\left(x^{2}-2x+1\right)\\ & = & {\left(1-x^{n}\right)}^{2}-n^{2}x^{n-1}{\left(1-x\right)}^{2}\\ & = & {\left(1-x\right)}^{2}\left[{\left(\sum_{i=0}^{n-1}x^{i}\right)}^{2}-nx^{n-1}\right]. \end{array}$$
Since
$${\left(\sum_{i=0}^{n-1}x^{i}\right)}^{2}=\sum_{i=0}^{n-2}\left(i+1\right)\left(x^{i}+x^{2n-2-i}\right)+nx^{n-1}$$
and
$$n^{2}x^{n-1}=\sum_{i=0}^{n-2}2\left(i+1\right)x^{n-1}+nx^{n-1},$$
we can rewrite the expression in the brackets of (Equation 20) as follows
$$\begin{array}{ccc} {\left(\sum_{i=0}^{n-1}x^{i}\right)}^{2}-nx^{n-1} & = & \sum_{i=0}^{n-2}\left(i+1\right)\left(x^{i}-2x^{n-1}+x^{2n-2-i}\right)\\ & = & \sum_{i=0}^{n-2}\left(i+1\right)x^{i}{\left(1-x^{n-1-i}\right)}^{2}>0. \end{array}$$
This shows that the ratios $\frac{M_{n+1}\left(x\right)}{M_{n}\left(x\right)}$, 0<x<1, are increasing for all n≥3, and $\frac{M_{n+1}\left(e^{-\alpha }\right)}{M_{n}\left(e^{-\alpha }\right)}$, n≥3, are decreasing with respect to α>0. Moreover,
$$M_{n}\left(e^{-\alpha }\right)=e^{-n\alpha }-ne^{-\alpha }+n-1=-n\alpha E_{\alpha }\left(Y_{1:n}-Y_{1}\right)>0.$$
We finally observe that
$$\begin{array}{ccc} B_{n}\left(\alpha \right) & = & -\frac{1}{n}\;\frac{M_{n}\left(e^{-\alpha }\right)}{D(1-e^{-\alpha })}=\frac{3}{n}\;\frac{M_{n}\left(e^{-\alpha }\right)}{M_{3}\left(e^{-\alpha }\right)}\;B_{3}\left(\alpha \right)\\ & = & \frac{3}{n}\;B_{3}\left(\alpha \right)\prod_{i=3}^{n-1}\frac{M_{i+1}\left(e^{-\alpha }\right)}{M_{i}\left(e^{-\alpha }\right)} \end{array}$$
is negative increasing, because so is $B_{3}\left(\alpha \right)$, and each $\frac{M_{i+1}\left(e^{-\alpha }\right)}{M_{i}\left(e^{-\alpha }\right)}$, i=3,…,n−1, is positive decreasing.Summing up, for every n≥2 we concluded that
$$\underset{\alpha >0}{\sup}B_{n}\left(\alpha \right)=\lim_{\alpha \to \infty }B_{n}\left(\alpha \right)=\lim_{x\to 0}\frac{1-n+nx-x^{n}}{2ex-1-x-x^{2}+x\ln x}=-\frac{n-1}{2}\;e.$$
Distribution functions (Equation 16) which attain the bound in the limit represent the mixtures of the exponential distributions with intensity α truncated right to the interval [0,1] with probability $1-e^{-\alpha }$, and atom at 1 with probability $e^{-\alpha }$. Making a standard location-scale transformation we obtain the distributions with arbitrarily chosen real μ and positive $\sigma_{1}$ which attain the bound in the limit. They are precisely described in the statement of Theorem 2. □

Theorem 2 provides a reliable estimate of the smallest possible time distance between the lifetimes of the single component and series system under the increasing failure rate assumption. Below we also determine more stringent bounds on the expectations of series system lifetimes under the more restrictive aging condition that the component lifetimes have increasing density functions.

**Theorem** **3** (ID distributions)**.***Suppose that the assumptions of Theorem* 1 *hold with*
p=1. *If*
n=2
*, then the bound*
(21)$$\frac{EX_{1:2}-\mu }{\sigma_{1}}\leq -\frac{1}{2},$$
*is sharp, and it is attained in the limit by the mixtures of uniform distributions on the interval*
$\left[\mu -\frac{2\sigma_{1}}{\alpha (2-\alpha )},\mu +\frac{2\sigma_{1}}{{\left(2-\alpha \right)}^{2}}\right]$
*with the atoms at*
$\mu +\frac{2\sigma_{1}}{{\left(2-\alpha \right)}^{2}}$
*when the probabilities of the atoms tend to* 1.
*For*
n≥3
*we have*
(22)$$\frac{EX_{1:n}-\mu }{\sigma_{1}}\leq -2\;\frac{n-1}{n+1}.$$
*If*
n=3
*, then the equality is attained by the mixture distributions described above for any fixed*
α∈(0,1]
*. For*
n≥4
*, the equality in* (Equation 22) *is attained by the uniform distribution on*
$\left[\mu -2\sigma_{1},\mu +2\sigma_{1}\right]$
*only.*


Bound (Equation 21) is identical with sharp bound for the lifetime of two-component system with arbitrary distributions of component lifetimes (see Goroncy [30], Corollaries 3.1 and 3.2). Surprisingly, bounds (Equation 22) coincide with the optimal evaluations for the expectation of the first order statistic based on an iid sample with symmetric unimodal parent distribution (see, Rychlik [31]).

**Proof.** We again refer to the norm maximization method. We can mimic the arguments of the proof of Theorem 2, with the only difference that instead of considering the elements of the space $L^{1}\left(R_{+},e^{-x}dx\right)$ we take into account the functions from $L^{1}\left(\left[0,1\right],dx\right)$. Consequently, we conclude that the possible candidates for quantile functions maximizing $\frac{EX_{1:n}-\mu }{\sigma_{1}}$ belong to the family of broken lines (Equation 2). The respective expectations of the single component and system lifetimes are presented in (Equation 3) and (4). Fixing p=1 in (Equation 6) we obtain the respective mean absolute deviation from the mean of the form
$$N_{1}\left(\alpha \right)=\frac{1}{4}\;\alpha {\left(2-\alpha \right)}^{2}.$$
This together with (Equation 5) give
(23)$$\frac{E_{\alpha }Y_{1:n}-EY_{1}}{N_{1}\left(\alpha \right)}=\frac{4}{n+1}\;\frac{1-\left(n+1\right)\alpha +\frac{n+1}{2}\alpha^{2}-{\left(1-\alpha \right)}^{n+1}}{\alpha^{2}{\left(2-\alpha \right)}^{2}}=C_{n}\left(\alpha \right),$$
say. The upper bound we look for amounts to the maximal value of the RHS of (Equation 23) with respect to 0<α≤1.We immediately check that $C_{3}\left(\alpha \right)=-1$, 0<α≤1. This means that for n=3, the upper bound is equal to −1, and it is attained by the arbitrary mixtures of standard uniform distributions with the atoms at 1. For other *n* we use the following result.Define functions
$$L_{n}\left(x\right)=2x^{n+1}-\left(n+1\right)x^{2}+n-1,\;0<x<1,\;n\geq 2,$$
with the derivatives
$$L_{n}^{\prime }\left(x\right)=2\left(n+1\right)\left(x^{n-1}-1\right).$$
We consider the ratios $\frac{L_{n}\left(x\right)}{L_{n-1}\left(x\right)}$ for n≥3. Their derivatives satisfy
$$\begin{array}{ccc} & & 2xL_{n-1}^{2}\left(x\right)\frac{d}{dx}\frac{L_{n}\left(x\right)}{L_{n-1}\left(x\right)}=\left(n+1\right)\left(x^{n-1}-1\right)\left[2x^{n}-nx^{2}+n-2\right]\\ & - & n\left(x^{n-2}-1\right)\left[2x^{n+1}-\left(n+1\right)x^{2}+n-1\right]\\ & = & 2x^{2n-1}-n\left(n-1\right)x^{n+1}+\left(n-2\right)\left(n+1\right)x^{n}\\ & + & \left(n-2\right)\left(n+1\right)x^{n-1}-n\left(n-1\right)x^{n-2}+2\\ & = & 2x^{n}\left(x^{n-1}-1\right)-2\left(x^{n-1}-1\right)-n\left(n-1\right)\left(x^{n}-x^{n-2}\right)\left(x-1\right)\\ & = & 2\left(1-x^{n-1}\right)\left(1-x^{n}\right)+n\left(n-1\right)x^{n}{\left(1-x\right)}^{2}\left(1+x\right)\\ & = & {\left(1-x\right)}^{2}\left[2\sum_{i=0}^{n-2}x_{i}\;\sum_{i=0}^{n-1}x_{i}-n\left(n-1\right)\left(x^{n-2}+x^{n-1}\right)\right]. \end{array}$$
The expression in the brackets can be rewritten as
$$\begin{array}{ccc} & & 2\sum_{i=0}^{n-2}\left(i+1\right)\left(x^{i}+x^{2n-3-i}\right)-n\left(n-1\right)\left(x^{n-2}+x^{n-1}\right)\\ & = & 2\sum_{i=0}^{n-2}\left(i+1\right)\left(x^{i}+x^{2n-3-i}-x^{n-2}-x^{n-1}\right)\\ & = & 2\sum_{i=0}^{n-2}\left(i+1\right)x^{i}\left(1-x^{n-1-i}\right)\left(1-x^{n-2-i}\right)>0. \end{array}$$
We concluded every fraction $\frac{L_{n}\left(x\right)}{L_{n-1}\left(x\right)}$, n≥3, is increasing on (0,1), and, in consequence, all $\frac{L_{n}\left(1-x\right)}{L_{n-1}\left(1-x\right)}$, n≥3, are decreasing there.Note that
$$C_{n}\left(\alpha \right)=-\frac{4}{n+1}\frac{L_{n}\left(1-\alpha \right)}{N_{1}\left(\alpha \right)}.$$
Obviously $N_{1}\left(\alpha \right)>0$. The same applies to all functions $L_{n}$, because
$$-\frac{L_{n}\left(1-\alpha \right)}{2(n+1)\alpha }=EY_{1:n}-EY_{1},$$
(comp. the first line of (Equation 5)), and the RHS is negative for all nondegenerate iid random variables. Also $\frac{L_{3}\left(1-\alpha \right)}{N_{1}\left(\alpha \right)}=1$ for all α. For n=2 function
$$C_{2}\left(\alpha \right)=-\frac{4}{3}\frac{L_{3}\left(1-\alpha \right)}{N_{1}\left(\alpha \right)}\frac{L_{2}\left(1-\alpha \right)}{L_{3}\left(1-\alpha \right)}=-\frac{4}{3}\frac{L_{2}\left(1-\alpha \right)}{L_{3}\left(1-\alpha \right)}$$
is decreasing, because the latter factor is positive increasing. Therefore
$$\underset{0<\alpha \leq 1}{\sup}C_{2}\left(\alpha \right)=\lim_{\alpha \to 0}C_{2}\left(\alpha \right)=-\frac{4}{3}\lim_{\alpha \to 0}\frac{-\frac{3}{2}\alpha +\alpha^{3}}{\alpha^{2}{\left(2-\alpha \right)}^{2}}=-\frac{1}{2}$$
(see (Equation 23)). This means that the bound for n=2 is equal to $-\frac{1}{2}$, and it is attained in the limit by the convex combinations of the uniform distribution on [0,1] and atom at 1 as the contribution of the continuous part decreases to 0.For n≥4 we have
$$C_{n}\left(\alpha \right)=-\frac{4}{n+1}\frac{L_{3}\left(1-\alpha \right)}{N_{1}\left(\alpha \right)}\prod_{i=4}^{n}\frac{L_{i}\left(1-\alpha \right)}{L_{i-1}\left(1-\alpha \right)}=-\frac{4}{n+1}\prod_{i=4}^{n}\frac{L_{i}\left(1-\alpha \right)}{L_{i-1}\left(1-\alpha \right)}.$$
The function is increasing because each factor of the product is positive decreasing, and attains its maximum $-2\frac{n-1}{n+1}$ at 1. This is the bound value and this is attained by the standard uniform distribution.The attainability conditions can be easily generalized to the families of distributions with arbitrary μ∈R and $\sigma_{1}>0$ if we replace (Equation 1) by the distribution functions of mixtures of uniform distributions on $\left[\mu -\frac{2\sigma_{1}}{\alpha (2-\alpha )},\mu +\frac{2\sigma_{1}}{{\left(2-\alpha \right)}^{2}}\right]$ with the atoms at $\mu +\frac{2\sigma_{1}}{{\left(2-\alpha \right)}^{2}}$. □

As we can predict, in the case of ID component lifetimes the negative bounds on the standardized expectations of series system lifetimes turn out to be more stringent than in the case of IFR components. Indeed, the absolute value of the right-hand side of (Equation 22) is $\frac{4}{e}\;\frac{n}{n+1}\approx 1.47152\frac{n}{n+1}$ times greater than its counterpart of (Equation 8). This shows that constructing systems of the component with either ID or merely IFR lifetime property significantly affects the system performance. Note that the bounds (Equation 8) and (Equation 22) as well as their ratio slightly depend on the size *n* of the series system.

## 4. Conclusions

In this paper we considered series systems whose components have independent and identically distributed lifetimes with either increasing failure rate and or increasing density function. The series systems are the simplest and most popular system used in technical applications, and the increase of failure rate or the density function of component lifetimes are natural analytic descriptions of the natural presumption that the components are wearing out during the system operation. It is obvious that the system lifetime is not longer than the lifetime of its any component. Here we studied the problem how much is the expectation of the system lifetime shorter than the expected lifetime of a single component under the assumption that the components have IFR and ID lifetimes. It occurs that the answer strongly depends on the units in which the difference is measured. In Section 2 we showed that the scale units based on the central absolute moments of component lifetimes with orders p>1 are too rough for detecting the differences, and the respective bounds expressed in these units provide only the trivial zero bounds. For detecting substantial differences, in Section 3 we used the mean absolute deviations from the mean units, and we determined sharp attainable strictly negative bounds on the expected differences between the system and component lifetimes under the IFR and ID assumptions. Theorems 2 and 3 show that that bounds under the more restrictive ID assumption are essentially less than those the the IFR case.

## Data Availability

Data sharing not applicable.

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
