# Peer review of "Bounds on the Lifetime Expectations of Series Systems with IFR Component Lifetimes"

_entropy, 2021, doi:10.3390/e23040385_

Round 1

Reviewer 1 Report

The paper deals with an issue of lifetime expectations of series systems consisting independent identical components. As to the formal aspects, the work is done well. All concepts are defined, all assumptions are listed and consecutive theorems are proved. The proofs are described in details. This is a strength of the paper. The reviewer tried to trace the described reasoning thread and was not able to find a mistake.

However, I feel that there is a lack of clear determination of the paper contribution. What is the novelty? How could be the work positioned in the relation to the state of art. Therefore, the Introduction section should be reworked. Please add a paragraph describing the contribution and the aim of your work and the structure of the paper.

Introduction section. You have cited 3 definitions of entropy. However, they are not referred in the article. Please clearly indicate in what way did you use this concept in your work directly or indirectly.

The paper ends without any discussion on the meaning of the obtained results. This is similarly as in the Introduction. The reader can ask: OK, I can agree. But what does it eventually mean? How can I take a benefit from this work?

So, it seems to me that somewhat like a conclusions section would be valuable. It would help a reader to embrace the contribution of the work. The Authors must be aware that scientific papers are often just briefly viewed by other researchers, who look for some ideas or solutions. The Authors should help such searchers to quickly determine the objectives of their own work.

Only a few references are newer than 5 years. Please add some recent references.

It seems the references are not in a desired format.

Author Response

Response to Reviewer 1:

We thank the referee for careful reading of the manuscript and valuable comments. The corrections suggested in the report were incorporated into the new version. All the modifications of the original text were distinguished with the red print. We also try to answer to all of the comments below.

„Lack of clear determination of the paper contribution, and lack of the conclusion section, and description of the paper structure”.:

The contribution is precisely determined in the abstract. To state it more distinctly, we added the conclusion section, and pointed out the novelty of our result in the introduction in the paragraph describing known bounds for lifetime expectations of k-out-of-n systems. We also included a small paragraph about the structure of the paper in the introduction.

„Some notions of entropy were presented in the introduction, and they were not referred in the further parts of the paper”.:

In our opinion, this is the problem that should be analyzed and judged by the editor. We were invited for contributing to the special issue by its guest editor Prof. Maria Longobardi who knew our research interests, and she was aware that we are not experts in the entropy theory. We wrote to her raising the problem again, and asking if possibly the paper on the bounds for the lifetimes of series systems with IFR components we were just completing possibly suits the scope of the issue. She answered positively, and encouraged us to submit this paper. We were also asked for pointing out some connections of entropy with the reliability systems and the IFR property of lifetime distributions. We mentioned them in the introduction. However, we should be fair and admit that the tools of the entropy theory were not used in our research, and we do not see direct applications of our results for the entropy studies. This was the reason we did not referred to the entropy notions except for the introduction section.

„Lack of new references, the reference list not in the desired format”.:

We included and briefly discussed several new references, and adapted the format of the references and the whole paper to the journal standards.

Reviewer 2 Report

The paper is mathematically correct and well detailed; the results are solid. It is globally well written, but I formulate the following comments for possible improvements, mainly on the overall motivation of the study:

o I dont understand the relation between the paragraph on entropy in Introduction and the main contributions of the paper. Please precise this relation since the paper is considered in the journal "Entropy". 

o In my opinion, the theorems lack of comment: Add comments and more explanations on their potential uses in theoretical or applied setting. Again, any link with the entropy concept is welcome. 

o A conclusion section is envisageable. 

Author Response

Response to Reviewer 2:

We thank the referee for careful reading of the manuscript and valuable comments. The corrections suggested in the report were incorporated into the new version. All the modifications of the original text were distinguished with the red print. We try to answer to all of the comments below.

„Precise the relation between entropy and the main contribution of the paper”:.

In our opinion, this is the problem that should be analyzed and judged by the editor. We were invited for contributing to the special issue by its guest editor Prof. Maria Longobardi who knew our research interests, and she was aware that we are not experts in the entropy theory. We wrote to her raising the problem again, and asking if possibly the paper on the bounds for the lifetimes of series systems with IFR components we were just completing possibly suits the scope of the issue. She answered positively, and encouraged us to submit this paper. We were also asked for pointing out some connections of the entropy theory with the reliability systems and the IFR property of lifetime distributions. We mentioned them in the introduction. However, we should be fair and admit that the tools of the entropy theory were not used in our research, and we do not see direct applications of our results for the entropy studies. This was the reason we did not referred to the entropy notions except for the introduction section.

„Lack on comments about the meaning of theorems”.:

Before formulating the main theorems we wrote some comments describing their meaning and practical significance.

„A conclusion section is envisageable”.:

We added the conclusion section.

Reviewer 3 Report

Cumulative residual entropy is an important concept and this manuscript is nicely taking it into account.

Better if the manuscript will be compiled using the mdpi (latex) template.

Other connections with Shannon entropy (10.3390/s20247346, 10.1515/auoc-2017-0006, 10.3390/e22121398) and order statistics (10.3390/econometrics9010001, 10.3390/s21010006, 10.3390/math8020216, 10.3390/e22111330) should be better covered in the introduction.

Also series systems have important applications in chemistry (10.1063/1.4918601).

Lines 28-30 - genetic algorithms and evolutionary theory find uses of series systems too (https://sciforum.net/paper/view/conference/3251). 

Also paper needs a summary and conclusion section.

Author Response

Response to Reviewer 3:

We thank the referee for careful reading of the manuscript and valuable comments. The corrections suggested in the report were incorporated into the new version. All the modifications of the original text were distinguished with the red print. We try to answer to all of the comments below.

„Compile the manuscript with the mdpi template”.:

We compiled the new version according to the journal standards.

Consider including several publications on connections of the Shannon entropy with order statistics, and applications of series systems in chemistry, genetics, and evolutionary theory”.:

We included and shortly discussed references suggested in the report. We thank the referee for drawing our attention to them.

„Paper needs a conclusion section”.:

We added the conclusion section at the end of the paper.

Round 2

Reviewer 1 Report

Since the Authors have addressed all my remarks, I have no further comments